# Prey killing without invasion by *Bdellovibrio bacteriovorus* defective for a MIDAS-family adhesin

Jess Tyson [1,4], Paul Radford [1], Carey Lambert [1,5], Rob Till[1,5], Simona G. Huwiler [2], Andrew L. Lovering [3] ✉ & R. Elizabeth Sockett [1] ✉

The bacterium *Bdellovibrio bacteriovorus* is a predator of other Gram-negative bacteria. The predator invades the prey's periplasm and modifies the prey's cell wall, forming a rounded killed prey, or bdelloplast, containing a live *B. bacteriovorus*. Redundancy in adhesive processes makes invasive mutants rare. Here, we identify a MIDAS adhesin family protein, Bd0875, that is expressed at the predator-prey invasive junction and is important for successful invasion of prey. A mutant strain lacking *bd0875* is still able to form round, dead bdelloplasts; however, 10% of the bdelloplasts do not contain *B. bacteriovorus*, indicative of an invasion defect. Bd0875 activity requires the conserved MIDAS motif, which is linked to catch-and-release activity of MIDAS proteins in other organisms. A proteomic analysis shows that the uninvaded bdelloplasts contain *B. bacteriovorus* proteins, which are likely secreted into the prey by the *Δbd0875* predator during an abortive invasion period. Thus, secretion of proteins into the prey seems to be sufficient for prey killing, even in the absence of a live predator inside the prey periplasm.

In their predatory lifecycle in liquid media, *Bdellovibrio bacteriovorus* predators swim rapidly to another Gram-negative bacterial prey surface and then attach to it, and some minutes later, proceed to invade into that prey bacterium's periplasm in a process involving Type IV pilus retraction[1–3]. The prey are killed and the predator consumes them and replicates within them. The initial predator-prey cell encounter of two live moving bacteria becoming attached brings a marked change in movement- speed of the predator. Also new associating and dissociating physical forces will occur between surface proteins as the predator interfaces with the initially-live prey cell surface[4].

The predator also squeezes through the outer envelope of the prey[2,5]; a process which also involves dynamic predator prey contacts. As intraperiplasmic replication inside a range of Gram-negative prey bacteria is a rapid mode of growth for *B. bacteriovorus* predators, evolution has equipped them to express many different surface adhesins to enter a wide range of Gram-negative prey. Recently studies of a family of 21 Mosaic Adhesive Trimeric (MAT) family adhesin proteins has shown that they are expressed at diverse sites on the predator surface, recognise many different prey epitopes, and spatially reorganise upon entry[6]. Due to the multiplicity of types, deletion of single MAT proteins only slightly delays what is otherwise wild type predation. This redundancy and diversity of predator adhesins is important and ensures that brief predator encounters with diverse prey do not fail in nature. As a result, non-invasive adhesin predator mutants are difficult to discover by simple mutagenesis and screening for predation defects. Our current study sheds new light on this topic.

A type of different protein adhesin system known in dynamic, force-involving, bacterial adhesion to eukaryotes and eukaryotic cell organelle activities is the MIDAS (metal-ion-dependent adhesion site) protein family. Garbarino and Gibbons discovered such integrin-like

[1]School of Life Sciences, University of Nottingham, Medical School, Queen's Medical Centre, Nottingham NG7 2UH, UK. [2]Department of Plant & Microbial Biology, University of Zurich, CH-, 8057 Zurich, Switzerland. [3]School of Biosciences, University of Birmingham, Birmingham B15 2TT, UK. [4]Present address: Chain Biotechnology Ltd, MediCity, D6 Thane Road, Nottingham NG90 6BH, UK. [5]Present address: Biodiscovery Institute, University of Nottingham, Coates Road, Nottingham NG7 2RD, UK. ✉e-mail: a.lovering@bham.ac.uk; liz.sockett@nottingham.ac.uk

proteins in 2002[7] by a genomic study, identifying their distinct AAA domain family and distant relationship to dynein, as indicative that MIDAS proteins may be adhesins involved in moving interactions at surfaces. Later research by Izore and co-workers in 2010[8] showed that *Streptococcus pneumoniae* uses a single MIDAS type adhesin to make contact with the extracellular matrix of eukaryotic cells. From genomic studies, they suggested that use of such an adhesin by Gram-positive bacteria to recognise eukaryotes may be widespread. They initially defined the MIDAS motif as Asp-X-Ser-X-Ser with an associated additional acidic residue. Mickolajaczyk and co-workers further examined the activities and functional processes of MIDAS domain proteins from eukaryotes, working in ribosome biogenesis[9]. They found that MIDAS protein binding time on substrates is enhanced by mild dissociative forces up to approximately 4pN and then that 10x stronger dissociative forces abolish binding. This and previous discoveries led to MIDAS proteins being seen as able to form catch bonds between two opposingly moving entities or substrates, analogous to the movements between predator and prey cells during bacterial predation. Signalling of attachment via a different accessory domain in the MIDAS proteins of eukaryotes were found to regulate other processes. One particularly pertinent example, involving both motility and invasion, analogous to predation[10], is the use of MIDAS adhesins by apicomplexan parasites. MIC2 from *Toxoplasma*[11] and the TRAP protein from *Plasmodium*[12] both use a MIDAS-containing (von Willebrand A) VWA domain to license stick-and-slip motility, from a non-engaged motor state to a motile state, on host cell ligands. Alternative roles for MIDAS proteins exist, most prominently as force sensors, which are not mutually exclusive with adhesion capability[13].

Here, we have discovered that a surface-expressed MIDAS protein Bd0875 is important for the invasively productive interaction between predatory *B. bacteriovorus* and bacterial prey, being expressed at the junction between invading predator and prey. Active site amino acids of its MIDAS motif are required for wild type predation. The Δ*bd0875* (and active MIDAS motif point mutant strains) gave a significant percentage of failed invasions. These resulted in formation of rounded empty bdelloplasts (killed bacterial prey which had been acted upon by the *B. bacteriovorus* Δ*bd0875* predator, but not invaded, distinct from wild-type predation). We found that a range of *B. bacteriovorus* proteins, components of a molecular kiss of death, were secreted into these empty bdelloplasts, even when invasion failed due to the Δ*bd0875* mutation. This MIDAS mutant phenotype is important, shedding light upon evolutionary scenarios of invasive *B. bacteriovorus* predation developing from a prior state of simple predator-prey attachment. *B. bacteriovorus* lives and replicates within bacterial prey cells in its wild type predatory lifestyle[14]. It also benefits from and uses prey-derived nutrients degraded, by enzymes it secretes, and which it consumes inside its prey's periplasm. Here however, we have discovered that a MIDAS deletion mutant can kill bacteria by effector secretion without the physical properties of whole cell invasion.

## Results and discussion
### Identifying Bd0875, a potential MIDAS adhesin and other B. bacteriovorus MIDAS proteins
At the start of our study, we aimed to discover new adhesins that may be involved in predatory invasion and to test them by gene deletion. A pilot preliminary experiment and published transcriptional analysis[14] were combined to identify the gene for MIDAS-motif potential adhesin protein Bd0875, and two other different MIDAS-motif containing proteins Bd0767 and Bd3132 as preliminary predation-associated candidates for comparative gene deletion functional studies. We chose one further MIDAS-motif potential adhesin protein Bd1483, which was more related to Bd0875 in structure (both include a lipobox motif, discussed below), but which was not found to be upregulated on contact with prey bacteria. Bd1483 was also included in the study as its

structure could be interrogated with reference to other characterised MIDAS adhesins.

In the pilot experiment an HI (host/prey -independently cultured) strain of *B. bacteriovorus* HD100 deleted for *bd1291* (itself identified as an operonal partner of the Type IVa pilus, (TFP), major pilin Bd1290) was found to show prolonged, stalled attachment, without invasion, on the surface of prey cells. Preliminary comparison of gene expression of this stalled invasion versus wild type HI invasion (which can still invade albeit at various rates; Supplementary Table 1), revealed three MIDAS motif genes, *bd0875* and also *bd0767* and *bd3132* which were upregulated in wild type upon contact with prey, but not in the stalled mutant. Genes (*bd0816* and *bd3459*) for two other previously characterised DacB proteins known to be secreted into prey and used during invasion[15] had a similar pattern of upregulated in wild-type, but not in the stalled mutant. Although the diverse rates of wild type HI invasions meant that this experiment could remain only a pilot, it revealed a testable candidate MIDAS adhesin, for *B. bacteriovorus*, as Bd0875 had a lipobox and was a potential outer membrane protein.

Interrogating previous transcriptional data[14] (from a full study of wild type predatory cultures which could be statistically tested), showed that *bd0875* was upregulated approximately 3-fold after mixing of wild-type predatory, host dependent *B. bacteriovorus* with prey (Supplementary Table 2). The genes encoding related proteins, all with an apparently related MIDAS fold Bd0767, Bd3132 and Bd1483 (explained below), were not similarly upregulated upon contact with prey. Of these four, Bd0875 and Bd1483 are the most similar, with an identifiable lipobox motif, the MIDAS fold inserted into a small beta-rich domain formed from the N and C termini, and both had been identified (along with Bd0767) in a *B. bacteriovorus* surface secretome study[16]. (The comparative structures are discussed in more detail later). Taken together, the contextual comparison of these transcription patterns implicated Bd0875 as an invasion-associated putative adhesin. Although protein Bd0875 resisted attempts at protein crystallization (at a time prior to Alphafold release), homology to Bd1483 and comparison to general MIDAS proteins allowed for precision mutagenesis of the MIDAS motif and other domains (discussed in more detail later). We were also able to include deletions of the more distally related Bd0767 and Bd3132 as comparators.

### MIDAS protein Bd0875 is important for correct prey invasion
We hypothesised that Bd0875 would be involved in dynamic adhesive encounters with prey bacteria and so studied it and the three other MIDAS comparators by deletion mutagenesis (Supplementary Table 3) and microscopic analysis of predation. We discovered that while all four deletion strains could be cultured predatorily on *E. coli* prey, the Δ*bd0767*, Δ*bd1483* and Δ*bd3132* mutants produced cultures with invaded prey bdelloplasts, indistinguishable from wild type HD100. Only the *B. bacteriovorus* Δ*bd0875* MIDAS mutant showed a predation defect (Fig. 1a; Supplementary Fig. 1), where although 90% of the prey bdelloplasts were invaded by predators; 10.1% (+/− 2.9%) of *E. coli* prey bdelloplasts from Δ*bd0875* predation were rounded but missing an invaded *B. bacteriovorus* cell. To the best of our knowledge, this aberrant invasion phenotype is unprecedented in the *B. bacteriovorus* literature.

We scored bdelloplast invasion using Bd0064-mCherry to cytoplasmically mark and visualise the Δ*bd0875 B. bacteriovorus* (Fig. 1a). The 10.1% (+/− 2.9%) of empty bdelloplasts persisted until beyond T = 300 min of the predatory cycle (Supplementary Fig. 2), well after normally invaded bdelloplasts had burst liberating *B. bacteriovorus* progeny. Live-dead staining with a propidium iodide-based stain showed that although no *B. bacteriovorus* had invaded the empty bdelloplasts the *E. coli* prey were dead (Fig. 1b; Supplementary Fig. 1), and they were also rounded in appearance. These empty bdelloplasts were smaller in area than wild type invaded bdelloplasts (Fig. 1b; Supplementary Table 4) (0.917 +/− 0.026 μm² Δ*bd0875* empty

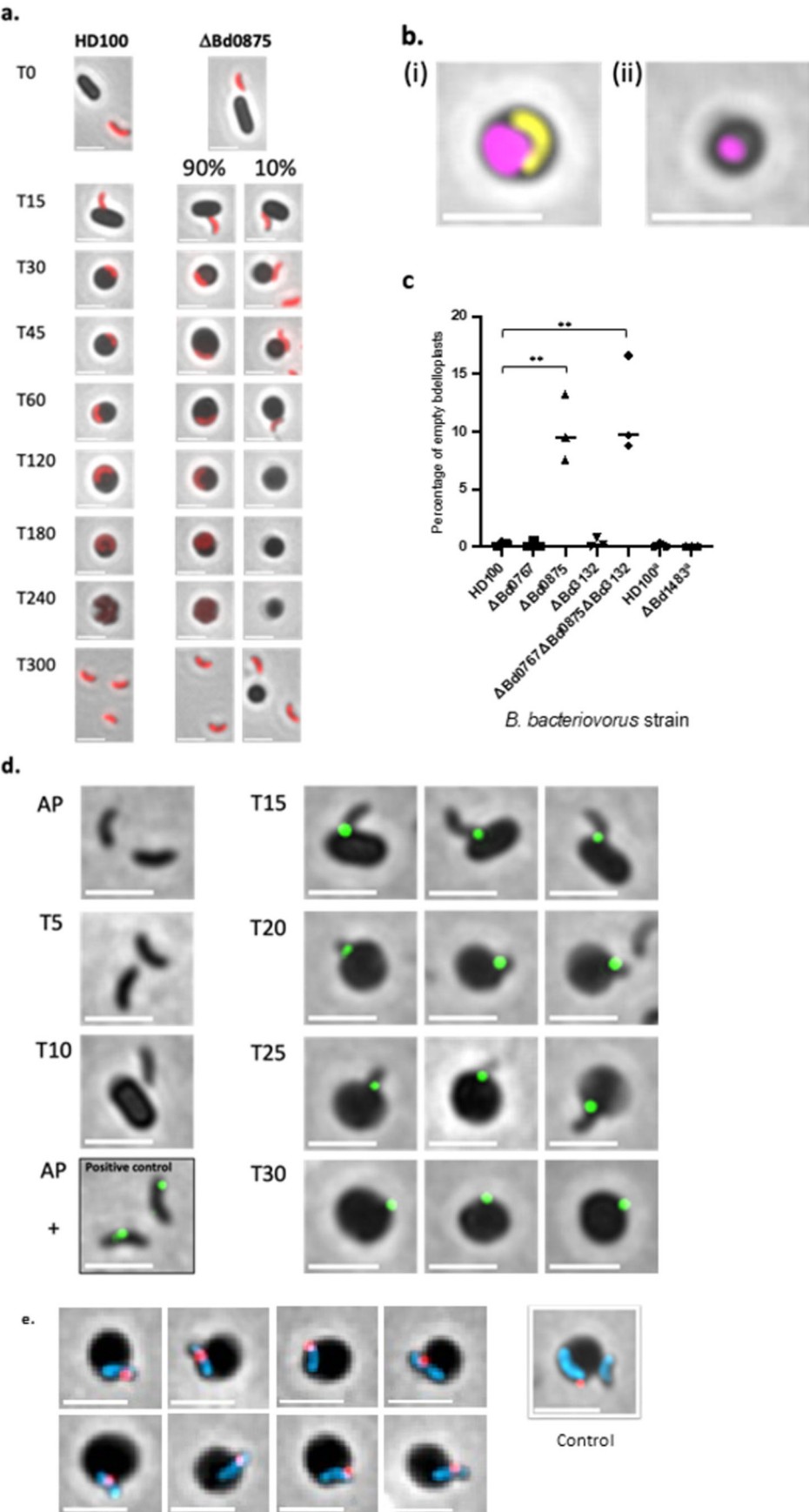

**Fig. 1 | Microscopy showing empty bdelloplasts of Δbd0875 predator and Bd0875 protein location. a** Microscopically imaged typical progress through predation, from attack phase (AP) outside prey to invasion, replication and exit; for cytoplasmically labelled *B. bacteriovorus* strains constitutively expressing Bd0064mCherry (red). Normal predation process (T0-T300 min after mixing predator and prey) occurs for HD100 wild type and 90% of Δbd0875 predators but 10% are devoid of an invaded of a Δbd0875 predator. Images from within one experiment and representative of three biological repeats **b** Examples of two types of bdelloplasts from live/dead staining of *E. coli* preyed upon by Δbd0875: (i) wild type predation with live (false coloured yellow) predator inside dead (false coloured magenta) prey (ii) empty dead (false coloured magenta) bdelloplasts without invaded predator, Scale bar 2 μm. Images representative of three biological repeats. **c** Frequency of dead empty *E. coli* prey bdelloplasts in predatory cultures of MIDAS gene deletion predator strains versus wild type HD100 monitored using live/dead staining. ᵃAnalysis of Δbd1483 was performed alongside a *B. bacteriovorus* HD100 control but in a separate experiment to the analysis of Δbd0767, Δbd0875, Δbd3132 and Δbd0767Δbd0875Δbd3132. Percentage of uninvaded bdelloplasts for Δbd0875 and Δbd0767Δbd0875Δbd3132 triple mutant were not significantly different from each other (p = 0 9041), but both of these mutant strains were significantly different from all other strains including versus HD100 (p values 0.0029 Δbd0875 and 0.0009 Δbd0767Δbd0875Δbd3132) (see Supplementary Tables 5a–c for full details). Data were derived from three biological repeats and over 1000 bdelloplasts per predator strain (Total and number of bdelloplasts counted per biological repeat are shown Supplementary Table 5d and as follows HD100 n = 1135, Δbd0767 n = 1181, Δbd0875 n = 1121, Δbd3132 n = 1223 Δbd0767Δbd0875Δbd3132 n = 1162; HD100ᵃ n = 1002 and Δbd1483 n = 1090. Mean (+SD) percentage empty

bdelloplasts for each strain is shown in Supplementary Table 5a. P values from one-way ANOVA are shown in Supplementary Table 5b and two-tailed unpaired t-test in Supplementary Table 5c. **d** Anti-mCherry (secondary antibody with Alexa Fluor 488 (green)) detection of positions of Bd0875-mCherry MIDAS adhesin protein expressed by a *B. bacteriovorus* HD100 strain chromosomally also WT for Bd0875. Fluorescence was detected on invaded prey (large, black) bdelloplasts and between invading, (small, black), predators and prey, but not on unattached predator at times of attack-phase (AP), (T5-10 min). A strain of *B. bacteriovorus* Bd2740mCherry MAT protein, with known surface expression during attack-phase (black outlined panel) was used as a Positive Control for detection of mCherry (green) in attack phase, even when Bd0875mCherry was not being expressed. No in vivo direct mCherry fluorescence from Bd0875mCherry was detected from the periplasm, hence surface antibody detection of mCherry was used here. Scale bar = 2 μm. Data derived from two biological repeats. **e** Typical range of different protein positions detected, at invasion junction, (as fluorescent predator is entering under the transparent outer-membrane of the prey), for Bd0875 adhesin proteins during predator invasion of prey (black) at T 25 min post mixing, using anti-mCherry (detected with secondary antibody with Alexa Fluor 555 (shown as red)). Bd0875-mCherry was expressed in *B. bacteriovorus* HD100 WT strain that was also cytoplasmically labelled with constitutively expressed cytoplasmic protein Bd0064mCerulean (blue) to illuminate the predator cell inside prey. An additional Control bdelloplast image, with contrasting externally attached (right hand side) and invaded (lefthand side) predators, is shown for comparison of invaded and just attached *B. bacteriovorus*. Scale bar = 2 μm. Range of protein positions illustrated from one biological repeat. Source data and full-sized images are provided in a Source data file accompanying this paper.

bdelloplasts versus 1.286 +/− 0.059 μm² for wild type-like-invaded bdelloplasts) but their circularity was similar to wild type (MicrobeJ circularity scores 0.989 +/− 0.001 for Δbd0875 empty bdelloplasts versus 0.986 +/− 0.003 for wild type-like invaded bdelloplasts). This is important as it was not previously known how *B. bacteriovorus* actually kill their prey. This result shows that no physical events caused by the predator being inside the prey inner periplasm are required to kill the prey. We had considered previously, apparently wrongly, that binding to the prey inner cytoplasmic membrane and/or disrupting proton motive force might cause death, because prey death came around the time of wild type predator full invasion[17]. However we previously found[18] a Δbd3279Δbd0468 double mutant could carry out predation in the outer periplasm, not in contact with the prey inner membrane, and in our current study, the idea that secreted agents kill the prey, rather than the physical, interior presence of predator, is reinforced.

Constructing and examining each of the selected MIDAS protein genes (*bd0767, bd1483, bd3132*) for any sign of empty bdelloplast production (Fig. 1c) showed that only the absence of Bd0875 gave rise to empty bdelloplasts.

## MIDAS adhesin Bd0875 locates at the moving predator-prey interface

C terminal-mCherry tagging of Bd0875 in *B. bacteriovorus* wild type HD100 (made as a single cross-over, so that both wild type Bd0875 and the mCherry-tagged version were expressed in these predators) produced a predatory culture with characteristics of wild type invasion. However, no in vivo Bd0875mCherry fluorescence was observed on the attack phase *B. bacteriovorus* themselves or inside the invaded prey bdelloplast at any time during predation. Testing with an anti-mCherry primary antibody, followed by detection with a fluorescent secondary antibody, versus a positive Bd2740mCherry MAT protein control[6] (Fig. 1d) gave no fluorescence on free-swimming attack phase Bd0875mCherry *B. bacteriovorus*, at timepoints T5 min and T10 min when not attached to prey. However, at T15 min when *B. bacteriovorus* were attached to prey by their non-flagellar pole, a focus of Bd0875-mcherry was detected (Fig. 1d) in 50% of *B. bacteriovorus* cells with 94% (n = 524) of the Bd0875-mcherry foci at the point of *B. bacteriovorus* attachment to the prey. The invading pole of *B. bacteriovorus* entered further into the prey; approximately halfway in at T20 min and three-

quarters in at T25 min. At these timepoints, the Bd0875-mCherry protein was detected at the predator-prey junction midway down the *B. bacteriovorus* at T20 min and then at the flagellar tail end of the *B. bacteriovorus* at T25 min as the last section of the predator cell entered the prey.

Finally, once the *B. bacteriovorus* had fully entered the prey, the Bd0875-mCherry was detected at T30 min as a point focus on the outside of the invaded *E. coli* prey bdelloplast.

This shows that the C terminus of Bd0875 is protruding from the predator at prey invasion as expected for a lipobox adhesin, likely lipid anchored from its N terminal region on the surface of the predator. Bd0875 adopts a dynamic position between the invading predator and the prey porthole[19] through which it enters; with the Bd0875 protein being left at the predator surface post prey entry. The dynamic range of Bd0875 adhesin positions at the predator-prey interface, during predator invasion of prey, was further demonstrated by staining the predator cytoplasm (Fig. 1e). We hypothesise that the dynamic activity of Bd0875, which is often seen on one side of the predator (Fig. 1d) could be acting to pivot the predator sideways (Fig. 1e) into the prey periplasm as it enters.

As we cannot express a lipobox-minus Bd0875 in a correctly secreted form, identification of the exact location of Bd0875 during predator invasion cannot be made (i.e., distinguishing between retention on predator or deposition on prey). However, the conserved lipobox at Cys22 and inferred lipidation of Bd0875 inform on these possibilities, suggesting that Bd0875 is limited to the predator outer membrane, acting as a dynamic adhesin during prey invasion (although we know that it could theoretically be proteolyzed during predation for transfer from predator to prey).

## *B. bacteriovorus* secreted predatory proteins are detected in empty bdelloplasts despite lack of Δbd0875 predator entry

To discover what the Δbd0875 mutant predators were doing to kill the *E. coli* that they failed to invade, we wanted to study empty bdelloplasts that persisted in Δbd0875 predatory cultures, at 300 min after predator-prey mixing, (when any invaded bdelloplasts had burst releasing attack phase predator). Rounds of density gradient centrifugation enriched for empty bdelloplasts away from attack phase Δbd0875 *B. bacteriovorus* with samples confirmed by live/dead staining

(Supplementary Fig. 3). Harvested empty bdelloplasts were sent for mass spectrometry, matching peptide hits to the *B. bacteriovorus* HD100 proteome (Supplementary Dataset 1), rather than solely to an *E. coli* prey one. We searched the *B. bacteriovorus* derived proteins in Supplementary Dataset 1 for those encoded by genes that were upregulated at 30 min of predation in our previous study[14] (Supplementary Table 6). This allowed us to ask whether proteins remained in the empty bdelloplasts that had been secreted there by predators early in the prey-recognition and invasion process (at 15–30 min) before the *B. bacteriovorus Δbd0875* predator fell off.

By the nature of the experiment (and the approximately 300 min that had elapsed from predator encounter with prey to empty bdelloplast harvesting), peptide numbers per *B. bacteriovorus* protein, in the *E. coli* uninvaded dead bdelloplasts, were not high (Supplementary Dataset 1 and Supplementary Table 6) but were a fingerprint of previous predatory secretion by the *Δbd0875* predator. There were also, as expected, some high-abundance non-predatory peptides identified (RNA polymerase, ribosomal proteins, flagellins) present due to bursting/breakage of some attack phase *Bdellovibrio* during centrifugation processes.

Even though our dead-empty bdelloplast proteomics uses prey harvested at 300 min after initial invasion (and so may miss secreted proteins that have been functional early in invasion but degraded by this time, which seems to be the case for DacB proteins Bd0816 and Bd3459 which round prey bdelloplasts), we detected (Supplementary Table 6) key secreted predatory proteins (peptidoglycan N-deacetylase Bd3279, and L,D transpeptidases Bd0886, Bd1176 which act on previously DacB- modified prey peptidoglycan crosslinks) that are known to mark and strengthen respectively, the prey cell wall at early timepoints (30 min after predator-prey mixing) during predator entry[18,19].

We also detected a large grouping of new potential prey-damaging proteins that have secretion signals and some of whose genes are upregulated transcriptionally at 30 min of predator invasion into prey (Supplementary Table 6). As a whole, these are likely to work as a significant consortium of prey damaging proteins, rather than containing one killer molecule, but together this project gives the field new predatory effectors to study.

The depth of this kiss of death secretome (Supplementary Dataset 1) prevents an exhaustive analysis here, but several members have an annotation that fits a bacterial prey-damaging, bdelloplast-forming role, and that importantly these often cluster into related functional subgroups. These include serine proteases (S1 grouping Bd1541, Bd2535, Bd2627; S8 grouping Bd0376, Bd2269, Bd2428, Bd2692), cysteine proteases (Bd0084, Bd1380, Bd3070) and Zn-peptidases (Bd0306). Other inferable hydrolytic enzymes include the wider AB-hydrolase family (Bd0718, Bd1012, Bd1192, Bd2811), phospholipase D-like enzymes (Bd1257, Bd1516, Bd3603), nucleases (Bd3586) and a glycosyl hydrolase (Bd1141). Alphafold predicts models for other secretome candidates in this dataset and future work will, we predict, reveal novel enzymatic activities/folds in this dataset.

## The MIDAS motif is key for Bd0875 function and avoidance of empty bdelloplast production

To be sure that the MIDAS activity of the WT Bd0875 itself was responsible for avoiding the empty bdelloplast phenotype seen in *Δbd0875* mutants, we made a PROMALS3D[20] structure guided alignment of Bd0875, using the related MIDAS protein Bd1483 and wider MIDAS structural homologs to reveal the conserved MIDAS motif (Fig. 2a). Both Bd0875 and Bd1483 have an identical domain architecture, with a lipobox leading into a small beta-rich domain that is comprised from the N- and C- terminal regions of the protein, with a recognisable MIDAS domain inserted within this such that the two domains fold largely independently of one another to form a structure resembling an L-shape (Fig. 2b).

Exploiting this, we designed *Δbd0875* complementation tests with wild type Bd0875 and variants encoded by point-mutated versions of the gene. These were: Bd0875-D66A, the key binding site aspartate on loop 1 of the conserved D-X-S-X-S MIDAS metal binding motif (Fig. 3ai, 3aii),(which co-ordinates active site metal, usually $Mg^{2+}$ ion binding)[21]. Bd0875-E213A, an acidic residue conserved across our four *B. bacteriovorus* MIDAS proteins (blue starred motif Fig. 2a, and equivalent to an acidic residue from loop 3 of the MIDAS fold that approaches the DXSXS motif. Similar acidic residues in other MIDAS proteins flank and are thought to sense ligand binding at the active site metal and may move when these proteins switch between bound/unbound states[8,21]. We also tested Bd0875-Y349A sited in the small domain of the adhesin distal and separate to the MIDAS region. These small domains vary in other bacterial MIDAS proteins and Tyr or Trp at this position is the most conserved non-structural residue. The domain could potentially communicate MIDAS site occupancy (akin to the small domains that flank the apicomplexan MIDAS domains[11,12]), but is not well understood in these bacterial domain architectures.

The dead, empty bdelloplast phenotype was fully complemented to wild-type invaded bdelloplasts by restoration of the wild type *bd0875* gene in the *Δbd0875* mutant strain and also by *bd0875* gene alone in the triple *Δbd0767Δbd0875Δbd3132* mutant strain (Fig. 3b; Supplementary Fig. 2).

Complementation tests in the *Δbd0875* mutant with Bd0875-Y349A fully restored wild type bdelloplast formation (Fig. 3c). However, both Bd0875-D66A and Bd0875-E213A did not complement the *Δbd0875* mutant with *bd0875*-D66A having no significant activity above the *Δbd0875* deletion level of empty bdelloplast formation, giving 7.5% +/− 1.2% empty bdelloplasts versus *Δbd0875* deletion giving 9.3% +/− 1.7%. The *bd0875*-E213A reduced with low significance the level of empty bdelloplasts to 5.6% +/− 1.8% (Supplementary Table 5h) (Fig. 3c). This shows that a consensus (active) MIDAS motif in Bd0875 is required for wild type prey entry and without it empty bdelloplasts form upon prey-entry failures.

To examine what was unusual about MIDAS protein Bd0875 and why only it was important for prey entry, we used our PROMALS3D[20] structure guided alignment of all four studied MIDAS proteins (Fig. 2a). This revealed two unique regions of Bd0875 sequence (pink spot labels Fig. 2a) missing from the others. These form a cap region above the MIDAS domain in Bd0875 only, as seen in a two-way comparison to the most similar lipobox-containing MIDAS protein Bd1483 (Fig. 2b). These unique capping domains likely confer the ability of Bd0875 to bind a novel partner or substrate, inferred from the position flanking the MIDAS pocket, not found in the other homologs. Although the cap represents a linear region of the Bd0875 fold (aa 143-176), it packs against two other loops (including the variable residues of the DXSXS motif and also several hydrophobic amino-acids which means it acts like a co-operatively folded element). For these reasons, simple cut-and-paste chimeras were not possible to construct and test. Likewise, the cap cannot be deleted/shortened as complementary hydrophobic amino-acids from elsewhere in the fold (at least 5 residues) would be exposed. Hence, we experimentally used the single amino-acid site directed mutagenesis of the MIDAS motif, discussed above, but were unable to test with chimeras or cap region deletion as the structural model shows they could abolish overall protein packing and folding. This structural novelty is likely responsible for the dynamic adhesive role of Bd0875 as predator encounters and invades through the prey outer membrane into the periplasm.

Bd0875 is conserved in *Bdellovibrio* that either enter or attach to bacterial prey to kill them (Supplementary Fig. 4). Both types interface tightly with the prey-periplasm through the OM, and Bd0875 is clearly important for such predator-prey encounters, and without it, for the invasive *B. bacteriovorus* HD100, a defect in prey invasion occurs.

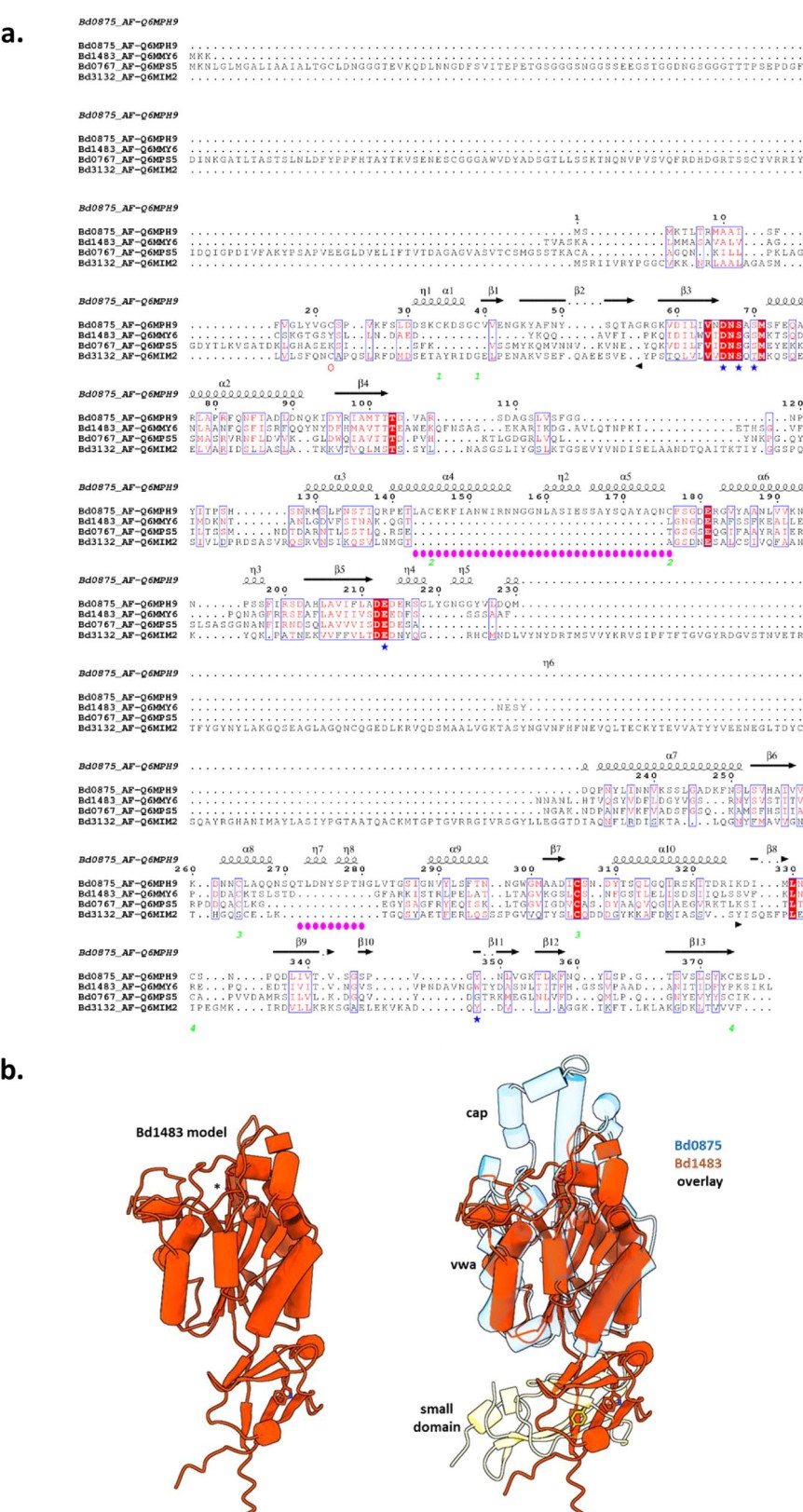

**Fig. 2 | Bd0875 protein has novel cap region absent in other *B. bacteriovorus* MIDAS proteins. a** PROMALS3D[20] structure-based sequence alignment of the four tested *B. bacteriovorus* proteins with MIDAS domain showing unique regions of Bd0875 protein region highlighted in pink; formatted using ESPRIPT[29]. **b** 3D alignment of MIDAS protein Bd1483 (Alphafold model), mapped onto the Bd0875 protein (Alphafold Q6MPH9and Q6MMY6); this illustrates the MIDAS adhesin domain (von Willebrand adhesin vwa) and that an additional cap region (pale blue) on the MIDAS domain (projecting upwards) is unique to Bd0875, the only MIDAS protein that gives a prey-invasion defect. Key to labelling: - The large run of pink circled amino acids in 2a corresponds to the main cap domain from 2b with a cap-adjacent small region encoded by the short run of pink circled amino-acids and the small domain (yellow in 2b) comprised of amino-acids 1-56 and 326-378 of Bd0875 sequence. Regions of alpha helices (α), beta sheets (β) and 3₁₀ helices (η), are annotated and numbered. The blue starred amino acids are those conserved in the MIDAS domain and the green numbered cysteines in the Bd0875 sequence are predicted to form disulphide bonds 1 to1, 2 to 2, 3 to 3, 4 to 4. Further Alphafold details are in Data Availability Statement.

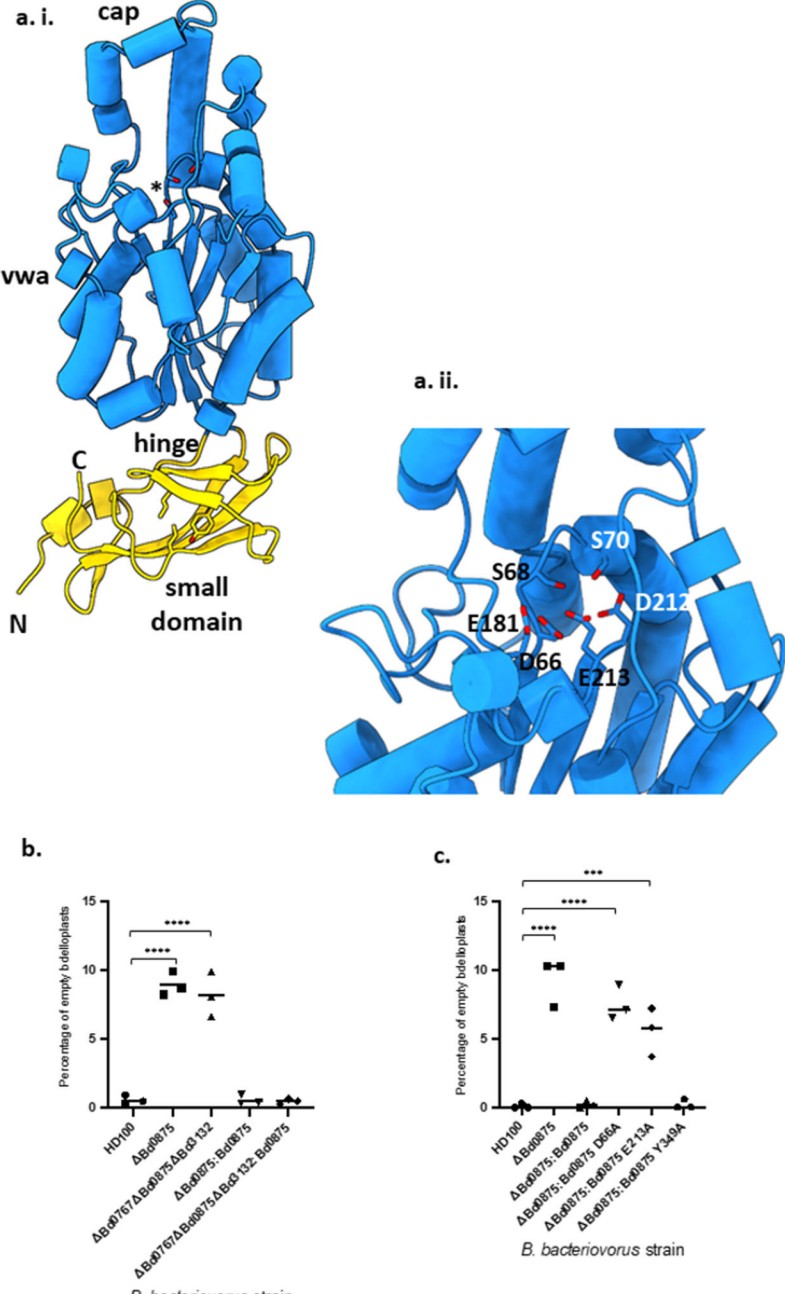

**Fig. 3 | Key MIDAS motif amino acids are required for Bd0875 protein function.**
**ai** Alphafold model of predicted Bd0875 structure (using Q6MMY6, signal peptide and lipobox of amino acids 1-22 omitted), showing Von-Willebrand adhesin (vwa) domain (blue), including Bd0875 cap region, plus hinge and small beta-rich domain formed from the N and C terminus of the Bd0875 protein sequence (yellow) with **aii** blue central domain showing numbered, conserved MIDAS motif amino acids on zoomed in adhesin structure; these are described and studied more in legend c. Further Alphafold details are in Data Availability Statement. **b** Percentage of dead empty *E. coli* prey bdelloplasts in cultures preyed upon by wild type HD100 and MIDAS gene deletion and gene complemented predator strains, verified using live/dead staining. (Supplementary Table 4) Mean (+SD) percentage empty bdelloplasts for each strain is shown in Supplementary Table 5e. *P* values from one-way ANOVA corrected for multiple comparisons are shown in Supplementary Table 5f. Total and number of bdelloplasts counted per biological repeat are shown in Supplementary Table 5g. Data for *bd0875* complementation of Δ*bd0875* and Δ*bd0767*Δ*bd0875*Δ*bd3132* triple mutant were not significantly different from each other (*p* > 0.9999) but each was significantly different from complementation tests and controls for all other strains (Supplementary Table 5f). Data were derived from three biological repeats and over 1000 bdelloplasts per predator strain. Total number of

bdelloplasts counted per biological repeat are shown supplementary Table 5g and as follows- HD100 *n* = 1181; Δ*bd0875 n* = 1151; Δ*bd0767*Δ*bd0875*Δ*bd3132 n* = 1354; Δ*bd0875*:*bd0875 n* = 1136 and Δ*bd0767*Δ*bd0875*Δ*bd3132*:*bd0875 n* = 1090. Source data are provided in a Source data file accompanying this paper. **c** Percentage of dead empty *E. coli* prey bdelloplasts in cultures preyed upon by wild type HD100 or MIDAS gene deletion strains undergoing *bd0875* complementation tests with wild type gene or point mutants of gene *bd0875* encoding: D66A MIDAS domain metal-binding active site, E213A MIDAS domain active site-proximal or Y349A small domain. *P* values from one-way ANOVA corrected for multiple comparisons are shown in Supplementary table 5i. HD100 control significantly different to data for deletion of Δ*bd0875* (*p* = <0.0001) and complementation of Δ*bd0875* with *bd0875*:D66A (*p* = <0.0001). For *bd0875*:E213A, data were significantly different to wild type HD100 (*p* = 0.0009). However *bd0875*:Y349A data were not significantly different to wild type HD100 (*p* = >0.9999). Data derived from three biological repeats and over 1000 bdelloplasts per strain as follows- HD100 *n* = 1002; ΔBd0875 *n* = 1014; ΔBd0875:Bd0875 *n* = 1187; ΔBd0875:Bd0875 D66A *n* = 1074; ΔBd0875:Bd0875 E213A *n* = 1073 and ΔBd0875:Bd0875 Y349A *n* = 1141. and shown in Supplementary Tables 5h-j where the significance of other comparisons can be viewed. Source data are provided in the Source data file.

Although we cannot experimentally quantify the adhesive role of Bd0875 at this point, (due to rapid and multiple predator arrival and departure events); Bd0875 could act as a pivot point, (being mostly detected expressed, Fig. 1d, on one side of the predator cell) helping as the *Bdellovibrio* enters the periplasm of the prey, augmented by other engines and adhesins such as TFP and MAT. This process may require a kind of turn, around the pivot point, by the predator because the periplasm is narrow and surrounds the prey centre (as discussed in a recent paper about vibroid predator cell shape and prey entry by Banks et al.[22]).

As Bd0875 is present on the surface of the invading *Bdellovibrio*, when motility engines are pulling the predator into the prey, we postulate that Bd0875 will be subject to forces that change its conformation and likely adjusts its binding activity in response.

We thus hypothesise that Bd0875 may be active as a kind of pivot point adhesin. The externally facing outer membrane predator surface location for Bd0875 adhesin detected in Fig. 1d, e is in keeping with that known for other lipobox containing MIDAS proteins in Gram negative bacteria and spirochaetes, including *Campylobacter* and *Haemophilus* adhesins[23]. This extracellular surface location for Bd0875 in *B. bacteriovorus* is also supported by the work of Dori-Bachash et al.[16] who also discovered that Bd0875 is part of a predicted extracellular/surface accessible proteome of HI-6 strain of *B. bacteriovorus*. This strain was grown prey independently but in a way that partially mimics predatory growth.

In summary, Bd0875 is a MIDAS adhesin with extra protein in the cap region above its MIDAS motif, and it is important for efficient bacterial prey invasion by the intraperiplasmic predator *B. bacteriovorus* HD100. Bd0875 protein has a lipobox and is expressed on the predator surface during prey invasion, located at the predator-prey junction site of dynamic cell-cell binding/unbinding and traction. We propose that the Bd0875 adhesin binds and rebinds predator to prey during dynamic prey-interaction and invasion processes.

This predator-prey location is relevant because MIDAS family proteins are known to perform catch and release binding activities in other cell systems[8–13]. We found that a functioning, consensus metal binding MIDAS motif is required for wild type prey invasion. In a Δbd0875 MIDAS deletion, or MIDAS active site D66A and E213A mutants, a significant proportion of prey invasions fail after a transient, abnormal interaction by the predator. However, despite this, the non-invasive Δbd0875 *B. bacteriovorus* mutant was found to secrete a potentially deadly cocktail of molecules, including previously proven predatory, prey-modifying proteins and a diversity of hydrolytic enzyme families, plus novel proteins, into the prey cells. These effectors round and kill prey, making empty bdelloplasts, even if the *B. bacteriovorus* mutant does not invade them in the usual way. This project potentiates future protein studies of ways to kill Gram-negative bacteria without full predatory invasion, a useful resource to the *Bdellovibrio* and antibacterial communities.

Our work on the Bd0875 MIDAS adhesin suggests that important catch and release adhesion events are occurring at the predator prey interface during invasion. We also firmly establish that it is not the physical presence of *B. bacteriovorus* inside the prey periplasm that is required for prey death, but that a combination of secreted predator molecules kill prey. Invasive predation gives wild type predators the benefit of access to the prey, as a safe haven, and food resource, which pays back the synthesis and secretion of the kiss of death mixture of predatory molecules.

## Methods

### Plasmid and strain construction

Primers used to generate fluorescently tagged, gene deletion and complementation strains of *B. bacteriovorus* are shown in Supplementary Table 3a, with the plasmid constructs in Supplementary Table 3b. Constructed strains are detailed in Supplementary table 3c.

### Bacterial strains and culturing

*B. bacteriovorus* HD100, gene deletion or fluorescently tagged strains were grown predatorily on stationary phase *E. coli* S17-1 prey in Ca/HEPES buffer (5.94 g/l HEPES free acid, 0.284 g/l Calcium Chloride dihydrate, pH 7.6) for 16 h, at 29 °C, with shaking at 200 rpm for liquid cultures, which were then filtered through a 0.45 μm filter to harvest *B. bacteriovorus* away from prey; or on YPSC overlay plates as plaques within a lawn of *E. coli* S17-1 prey[24]. To provide selection for fluorescent tags a final concentration of 50 μg/ml of kanamycin sulphate was added to growth media where appropriate. *E. coli* S17-1 cells were grown for 16 h in YT broth at 37 °C with shaking at 200 rpm.

### Generating gene deletion mutants in *B. bacteriovorus*

Single gene deletions of *bd0767*, *bd0875*, *bd1483* and *bd3132* (and deletion of *bd1291* for the strain used in the pilot experiment) from *B. bacteriovorus* HD100 were achieved by using the GeneArt Gibson assembly kit (ThermoFisher A46624) for Gibson assembly of the gene deletion construct into a suicide plasmid pK18mobsacB[25] and double reciprocal recombination into the predator chromosome (Supplementary Table 3). To construct *B. bacteriovorus* HD100 strains containing deletion of multiple genes, sequential deletion using the single/double deletion strain as the recipient strain, was used to construct the three gene deletion strain *B. bacteriovorus* HD100:Δbd0767Δbd0875Δbd3132 (Supplementary Table 3abc). All deletion strains were confirmed by PCR and Sanger sequencing. To visualise any effect of the single gene deletion and gene complemented strains during the predatory cycle, a C-terminal mCherry fusion to Bd0064 was introduced via single-crossover recombination into the predator genome (Supplementary Table 3abc)[22,26].

Strains for complementation tests were constructed by gene replacement, recombining the gene/variant under test by double crossover from pK18mobsacB to replace the Δbd0875 deletion in the genome. Constructs for complementation tests were verified by Sanger sequencing. Plasmid p0875_comp (Supplementary Table 3b) was introduced into *B. bacteriovorus* HD100Δbd0875 and HD100:Δbd0767Δbd0875Δbd3132 by conjugation (using donor *E. coli* S17-1 strains) and subsequently cured of the donor plasmid by sucrose suicide counter-selection, resulting in the integration of the gene complementation constructs via double crossover homologous recombination. To generate point mutated variants for complementation tests pK18bd0875:Asp66Ala, pK18bd0875:Glu213Ala, and pK18bd0875:Tyr349Ala (Supplementary Table 3b), point mutations were generated using a Phusion Site Directed Mutagenesis Kit (ThermoFisher Cat #F541) and pK18mobsacB containing *bd0875* as a template for subsequent conjugation into HD100 Δbd0875. Full integration of each gene/variant into the *B. bacteriovorus* genome was confirmed by PCR using flanking primers and sequencing.

### Microscopy timecourse of predation analysis

*B. bacteriovorus* wild type HD100:Bd0064mCherry (single cross over), deletion mutant Δbd0875:Bd0064mcherry (single cross over)and complemented strain Δbd0875: bd0875 Bd0064mcherry (single cross over) were cultured for 16 h as described above. Approximately synchronous predation of *E. coli* S17-1 pZMR100 by *B. bacteriovorus* strains was prepared by combining a 0.45 μM filtered, 10x concentrated *B. bacteriovorus* predatory culture with *E. coli* S17-1 pZMR100 (standardised to OD600 of 1) and Ca/HEPES at a ratio of 5:4:3, respectively[6].

Progress through the predatory lifecycle was visualised via fluorescence microscopy at 0, 15, 30, 45, 60, 120, 180, 240 and 300 min post mixing of predator and prey, by withdrawing 10 μl of the predatory culture and immobilising on a thin 1% Ca/HEPES buffer agarose pad and rapidly visualising this.

Adding the liquid culture to prepared agarose slides does not halt predation, but imaging all the views for each sample takes less than 5 min and we controlled for this by imaging and saving the

comparisons as wild type first in 2 experiments and the mutant first in 2 experiments. We then moved on to the next sampled timepoint for both cultures repeating the process as described. All cultures and prepared slides were adjacent to the microscope in our lab to facilitate this speed with timers running. All the analysis was done after the event using Fiji Version 1.52n and Microbe J version 5.13j, so that fields of view were focussed and imaged as quickly as possible.

Cells were visualised with a Nikon Ti-E inverted epifluorescence microscope and images acquired on an Andor Neo sCMOS camera with Nikon NIS software version AR5.11.02 64 bit[22]. Images were analysed using ImageJ[22,27] with minimal processing using the sharpen and smooth tools and adjustments to brightness and contrast to ensure clarity.

### Synchronous cultures for live/dead staining

Bacterial viability was tested microscopically by live-dead staining using LIVE/DEAD BacLight Bacterial Viability Kit (Molecular probes, Invitrogen). Synchronous predatory cultures were set up as above. At 120 min post mixing of predator and prey, 20 μl of predatory culture was removed and 0.1 μl mix of equal volumes of SYTO9 and propidium iodide added. 10 μl of this was immobilized on a thin agarose pad consisting of 1% agarose Ca/HEPES buffer and visualised with a Nikon Ti-E inverted epifluorescence microscope. Syto-9 and propidium iodide-stained cells were simultaneously viewed and imaged in the GFP channel (excitation: 470 nm, emission: 515 nm) and mCherry channel (excitation: 555 nm, emission: 620–660 nm) respectively. Images were obtained from three biological repeats. Full fields of view were visually scored for complete *B. bacteriovorus* entry or no entry into prey, along with Live/Dead colouring and percentage of bdelloplasts devoid of *B. bacteriovorus* calculated. Statistical analysis was performed in Prism 8.2.0 (GraphPad) and Excel 2016. Data were first tested for normality and then analysed using two-tailed unpaired *t*-test. The morphology and size of bdelloplasts were measured using the MicrobeJ plugin[28] for ImageJ as described[22]. False colouring of magenta (for dead red) and yellow (for live green) was applied to make Fig. 1b more easily visible (see Source Data file).

### Fluorescence tagging and microscopic detection of *B. bacteriovorus* Bd0875

To visualise expression of Bd0875 throughout the initial stages of predation, the mCherry tag was fused to the C-terminus of Bd0875 by PCR amplification of the gene, without its stop codon, and amplification of the *mCherry* gene. These two PCR products were assembled into pK18*mobsacB*[25] using the GeneArt Gibson assembly kit (Thermo-Fisher A46624) according to manufacturer's instructions. Each construct was conjugated into *B. bacteriovorus* HD100[22]. Since no direct fluorescent signal was observed, visualisation of external mCherry-tagged Bd0875 was performed by setting up synchronous predatory cultures as above, and removing aliquots every 5 min in the initial stages of predation and labelling via a rabbit anti-mCherry polyclonal antibody (Invitrogen PA5-34974) and detection with a secondary antibody of goat anti-Rabbit IgG with Alexa Fluor 488 (green) (Invitrogen A32731) for *B. bacteriovorus* strain WT for Bd0875 and additionally expressing C- terminally tagged Bd0875mCherry, Fig. 1d). Secondary antibody of goat anti-Rabbit IgG with Alexa Fluor 555 (orange shown as red in Fig. 1e) (Invitrogen A32732) was used for a *B. bacteriovorus* strain cytoplasmically labelled with constitutively expressed PilZ protein Bd0064mCerulean and additionally expressing C- terminally tagged Bd0875mCherry as described[6]. All antibodies were used a 1:1000 dilution. 10 μl aliquots were then immobilized on a thin agarose pad consisting of 1% agarose Ca/HEPES buffer and cells visualised with a Nikon Ti-E inverted epifluorescence microscope[22] using the following filters: GFP (excitation: 470 nm, emission: 515 nm) for Fig. 1d; or for Fig. 1e CFP (mCerulean; excitation: 440 nm, emission: 470–490 nm) and mCherry channel (excitation: 555 nm, emission: 620–660 nm) respectively. Images were minimally processed using sharpen and smooth tools (ImageJ[27]) with adjustments to brightness and contrast to ensure clarity. Full fields were visually inspected, and location of fluorescent foci noted post mixing of predator and prey.

### Enrichment of empty bdelloplasts by Percoll gradient centrifugation

To prepare and analyse empty bdelloplasts two predatory cultures of *B. bacteriovorus Δbd0875* (each 1 L Ca/HEPES buffer + 60 ml overnight culture of *E. coli* S17-1 + 50 ml of predatory culture of *Δbd0875 B. bacteriovorus*) were prepared and incubated 16 h with 200 rpm shaking at 29 °C and then filtered through a 0.45 μm filter. These cultures were 10-fold concentrated by centrifugation to make inocula for synchronously predatory cultures.

Three replicate, approximately synchronous predatory cultures of *E. coli* S17-1 preyed upon by *B. bacteriovorus Δbd0875* were prepared by combining the 0.45 μM filtered, 10x concentrated *B. bacteriovorus* predatory culture with *E. coli* S17-1 (standardised to OD600 of 1.0) and Ca/HEPES at a ratio of 5:4:3 respectively, in a final volume of 120 ml Ca/HEPES buffer in 25 ml flasks with incubation as before (x 3). Predation by *B. bacteriovorus Δbd0875* was allowed to continue for 5 h so that any normally invaded bdelloplasts (90% of the culture) were formed and consumed (at approx. 4 h) by the invaded predator. This left the approximately 10% empty but dead bdelloplasts remaining in the culture and an excess of attack phase *Δbd0875 B. bacteriovorus* and some lysed prey debris. These had different shapes and buoyant densities from each other. All cells and empty bdelloplasts were then harvested by centrifugation. To concentrate and pool material, the three 120 ml cultures were pooled and split across eight 50 ml falcon tubes for initial centrifugation at $3877 \times g$ for 20 min. Pellets were resuspended and centrifuged at $2000 \times g$ for 5 min before a final resuspension and pooling in 10 ml Ca/HEPES buffer. Samples were mixed with Percoll to 60% (v/v) in 30 ml OakRidge tubes and centrifuged in an A841 angle rotor in a Sorvall Ultracentrifuge at $8000 \times g$ for 30 min, which concentrated a band of empty bdelloplasts which was harvested and reapplied to a new gradient. This process was repeated 6 times first four times with 60% Percoll (v/v with Ca/HEPES buffer), then twice with 50% Percoll. Each time after spinning to form a gradient, a band formed near the top of the gradient enriched for empty bdelloplasts whilst *B. bacteriovorus* attack phase cells pelleted at the bottom of the gradient and broken prey debris was spread down the gradient. This top band was retrieved and applied to the subsequent Percoll gradients and centrifuged again for further enrichment. The final retrieved band containing the cleaner empty bdelloplasts (Supplementary Fig. 3 (vi)) was made up to 14 ml, diluting out the Percoll by adding water and this was centrifuged for 10 min at $5500 \times g$ to concentrate and pellet the empty bdelloplasts and remove the Percoll retained in the supernatant. The pellet was resuspended in 200 μl Ca/HEPES buffer and aliquots of 50 μl were stored at −80 °C.

### Proteomic analysis of *B. bacteriovorus* proteins in empty bdelloplast contents

Proteomic analyses were carried out on one 50 μl aliquot of the empty bdelloplasts (purified in the section above), by Oxford University Advanced Proteomics (https://www.proteomics.ox.ac.uk/). Protein samples were denatured with 4 M urea in ammonium bicarbonate buffer (100 mM) for 10 min at room temperature. After denaturation, cysteines were reduced with of TCEP (10 mM) for 30 min at room temperature and alkylated with 2-Chloroacetamide (50 mM) for 30 min at room temperature in the dark. Samples were then pre-digested with LysC (1 μg/100 μg of sample) for 2 h at 37 °C. Before overnight digestion with trypsin (1 μg/40 μg of sample) at 37 °C, urea was diluted down to 2 M in ammonium bicarbonate buffer (100 mM) and calcium chloride was added at 2 mM final. The next day, tryptic digestion was stopped with the addition of formic acid (5%). Digested

peptides were centrifuged for 30 min at 13,200 rpm at 4 °C to remove undigested material and aggregates. The digested peptides contained in the supernatant were desalted onto hand-made C18 stage tips before LC-MS/MS analysis pre-activated with 100% acetonitrile by centrifugation at 4000 rpm at room temperature. Peptides were washed twice in TFA 0.1%, eluted in 50% acetonitrile/0.1% TFA and speed-vacuum dried. Dried peptides were resuspended into 5% acetonitrile / 5% formic acid before LC-MS/MS analysis. Peptides were separated by nano-liquid chromatography (Thermo Scientific Ultimate RSLC 3000) coupled in line a QExactive mass spectrometer equipped with an Easy-Spray source (Thermo Fischer Scientific). Peptides were trapped onto a C18 PepMac100 precolumn (300 μm i.d.x5 mm, 100 Å, ThermoFischer Scientific) using Solvent A (0.1% Formic acid, HPLC grade water). The peptides were further separated onto an Easy-Spray RSLC C18 column (75um i.d., 50 cm length, Thermo Fischer Scientific) using 120 min linear gradient (15% to 35% solvent B (0.1% formic acid in acetonitrile)) at a flow rate 200 nl/min. The raw data were acquired on the mass spectrometer in a data-dependent acquisition mode (DDA). Full-scan MS spectra were acquired in the Orbitrap (Scan range 350–1500 m/z, resolution 70,000; AGC target, 3e6, maximum injection time, 100 ms). The 20 most intense peaks were selected for higher-energy collision dissociation (HCD) fragmentation at 30% of normalized collision energy. HCD spectra were acquired in the Orbitrap at resolution 17,500, AGC target 5e4, maximum injection time of 120 ms with fixed mass at 180 m/z. Charge exclusion was selected for unassigned and 1+ ions. The dynamic exclusion was set to 40 s. Tandem mass spectra were searched using Sequest HT in Proteome discoverer software version 1.4 against a protein sequence database containing 8223 protein entries, including *Bdellovibrio bacteriovorus* HD100, *Escherichia coli* K12 and contaminant proteins. During database searching cysteines (C) were considered to be fully carbamidomethylated (+ 57,0215, statically added), methionine (M) to be fully oxidised (+ 15,9949, dynamically added), all N-terminal residues to be acetylated (+ 42,0106, dynamically added). Two missed cleavages were permitted. Peptide mass tolerance was set at 50ppm on the precursor and 0.6 Da on the fragment ions. Data was filtered at FDR below 5% at PSM level.

Proteins identified in the empty bdelloplasts are presented in supplementary data (Supplementary Dataset 1), with a tab displaying all peptides identified (including the prey peptides) and a tab with the *B. bacteriovorus* peptides. We also display the *B. bacteriovorus* peptides (Supplementary Table 6) sorted by expression of the genes encoding them levels 30 min predation from a previous study[14]. This highlights potentially predatory proteins whose gene expression is induced by invading *B. bacteriovorus* and not in attack phase *B. bacteriovorus* outside prey[14]. While we appreciate that further analyses can and should be carried out on these proteins they represent candidates of interest for further studies beyond this work.

### Reporting summary

Further information on research design is available in the Nature Portfolio Reporting Summary linked to this article.

## Data availability

Source data for the Figures and statistical analyses are presented in the Source Data file. Transcriptomic data for *B. bacteriovorus bd1291* deletion mutant and wild type HD100 strain are deposited at the (Genbank) Sequence Read Archive (SRA) with accession numbers SAMN38260227 to SAMN38260240. The previously published transcriptional data[13] of wild type predatory *B. bacteriovorus* HD100 upon invasion of prey, used to compare to the proteomics in Supplementary Dataset 1, are available at GEO Gene expression Omnibus GSE9269. The AlphaFold models of Bd0875 and Bd1483 proteins were from DB version 2022-11-01, created with the AlphaFold Monomer v2.0 pipeline (accessed at https://alphafold.ebi.ac.uk/entry/Q6MPH9

and https://alphafold.ebi.ac.uk/entry/Q6MMY6, respectively). The mass spectrometry proteomics data have been deposited to the ProteomeXchange Consortium via the PRIDE[30] partner repository with the dataset identifier PXD050423 and are also presented in Supplementary Dataset 1 of this paper. Source data are provided with this paper.

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

## Acknowledgements

J.T., P.R., C.L., R.T., A.L.L., and R.E.S. were funded by the Wellcome Trust Investigator Award in Science (209437/Z/17/Z) (R.E.S. and A.L.L.). S.G.H. is funded by the Swiss National Science Foundation (PZ00P3_193401) (S.G.H.). For the purposes of open access, the corresponding author has applied a CC BY public copyright licence to the Author Accepted Manuscript version arising from this submission. We thank Marjorie Fournier and Aygul Malone from the Oxford Advanced Proteomics Facility (https://www.proteomics.ox.ac.uk/) for proteomic analysis. R.E.S. would like to thank her collaborators and the *Bdellovibrio* community for support and interest in our lab's research, as she closes her lab for family reasons.

## Author contributions

R.E.S. and A.L.L. jointly conceived the project and obtained funding. J.T. discovered empty bdelloplast MIDAS adhesin phenotype, carried out fluorescently tagged and live/dead microscopic analysis and quantitation of phenotypes for all mutants and tagged strains, tested protein location microscopically, with antisera, grew large scale cultures and analysed fractions from centrifugation for proteomics preparations of empty bdelloplasts, constructed and verified some of the fluorescent strains used. P.R. designed, constructed, and verified the majority of deletion, point mutated and fluorescent strains, C.L. carried out pilot transcriptomic analysis of Bd1291 strain and other transcriptomic analyses, purified empty bdelloplasts by Percoll gradient centrifugation for proteomic analysis and analysed proteomic results with respect to transcriptional timing of previously published gene expression at prey invasion. R.T. constructed some of the deletion strains for the study. S.G.H. helped with transcriptional analysis of gene expression during predation throughout the manuscript. A.L.L. identified the MIDAS proteins in *bd1291* mutant transcriptional data, carried out protein structural comparison and analysis and selected point mutants for construction. R.E.S. supervised and suggested lab experimentation and helped analyse data. J.T., R.E.S., and A.L.L. wrote the manuscript with data and analysis from P.R., C.L., and S.G.H. and editing from authors. All authors approved of the final version before submission.

## Competing interests

The authors declare no competing interests.
