## [Peer Review File · Nature Communications]

Prey killing without invasion by *Bdellovibrio bacteriovorus* defective for a MIDAS-family adhesin.Reviewer #1 (Remarks to the Author):

The authors describe a very interesting phenotype in the predatory bacterium *Bdellovibrio bacteriovorus*. This work is significant for multiple topic areas, including bacterial predation, the overall roles of adhesins in bacteria, and evolution and ecology of bacterial lifestyles. This work advances our understanding of the role of specific proteins in the stages of *Bdellovibrio* predation, which is an important area of focus with potential practical applications against bacterial pathogens.

Please see specific comments below -

Line 19:

Bd0875..."important for efficient attachment to and successful invasion of prey"

The authors clearly demonstrated the role of Bd0875 in successful invasion of prey, but I'm not convinced about the role of Bd0875 in efficient attachment to prey. The experiments here focused on invasion, specifically the presence of empty but still rounded bdelloplasts as an indicator of failed invasions. To me, the current version of the manuscript doesn't support a conclusion about the role of Bd0875 in attachment.

Lines 36-38:

the dynamics of the initial interaction between predator and prey is a fascinating topic, but I'm not sure what this sentence is meant to convey about this interaction.

"marked change in movement-speed of the predator" - is this describing the predator's approach to a potential prey cell prior to attachment?

"a struggle of associating and dissociating forces between the two initially live motile bacteria" - is this describing the physics of predators interfacing with prey, or molecular interactions between predator and prey outer surface components?

I suggest expanding explanations here and possibly including a citation.

Line 40:

"contracts" should be "contacts"?

Line 74:

"license 'stick-and-slip' motility on host cell ligands"

I'm not sure what "license" means in this context

Line 82:

"gave a significant percentage of failed predations"

are you considering these failed predations because the prey is killed but the predator does not replicate? does the predator gain any nutrients from killed but empty bdelloplasts? what is the fate of the Bd0875 mutant predator that attaches, kills, but fails to invade?

Line 108:

what was the "parent strain" for the HI Bd1291 mutant? was it HD100?

Line 119:

"for *B. bacteriovorus* as Bd0875 has a lipobox" - delete "as"?

Lines 304-306:

"AlphaFold models of other secretive candidates are strongly suggestive of many novel enzymatic activities/folds in this dataset"

support for this statement?

Figure 2a:

the following information would be helpful to the reader either in the legend or near the first mention of this figure in the text -

what do the green numbers indicate? what do the blue stars indicate?

there are two regions marked with pink circles - does the longer one correspond to the cap, and the smaller correspond to the "small domain" shown in yellow in the structure overlay? or do both

regions form the cap?

Line 404:

"a significant proportion of prey invasions fail after a transient, abnormal attachment of the predator" - can you include or explain some data to show how attachment is transient and abnormal compared to wild-type?

a general comment:

is Bd0875 found in other intra-periplasmic Bdellovibrio? if so, are the key features conserved?

Reviewer #2 (Remarks to the Author):

In this manuscript, Tyson and coworkers set out to determine the importance of a key step in predation of prey bacteria by *Bdellovibrio bacteriovorus*: prey entry. By combining proteomics, microscopy, and protein structural modeling, they were able to uncover a process they term the "kiss of death" where prey cells round up and die forming empty bdelloplasts, but the predator remains outside the prey cell. This process occurs in about 10% of bdelloplasts invaded by a strain of *B. bacteriovorus* lacking Bd0875. By synchronized imaging of predation, predator cells lacking Bd0875 attach to, but do not enter, the prey cells. The authors go on to examine the structural features of Bd0875 and structurally similar MIDAS proteins (Bd0767, Bd1483, and Bd3132). They use a genetic complementation approach to show that MIDAS domain was the key component to prevent empty bdelloplast formation. In addition, the authors were able to harvest empty bdelloplasts through density gradient centrifugation and determine that this "kiss of death" includes secretion of key predatory proteins. In general, the data presented mostly support the claims listed in the manuscript. I have included a few areas below where the authors should expand their experiments and some minor textual enhancements.

1) Microscopy sampling and the stopping of predation. The authors should be complimented that they have such exquisite temporal control over their synchronized predation time courses.

1A) It is confusing to me if the process of sampling 10 μ L of culture and immobilizing on a thin agarose pad is able to immediately halt predation. Given the dynamic nature of the process, the authors should comment on how they control for time since immobilization, how long it takes from the sampling of the predation flask to capturing the fluorescent images, and how long it takes to capture all the images.

1B) The images chosen for figure 1d portray a compelling story where the predator cell is able to enter the prey cell. However, I am not sure I am convinced that these images necessarily imply the predator has entered, and rather could be sitting on the surface of the bdelloplast. It may be that the authors know from personal experience, or different focal planes, or from dynamic/movies that the difference of internalized and attached but external are obvious, but this has not been presented clearly in this manuscript.

2) Metal binding and MIDAS function.

2A) As Bd0875 expression is upregulated in in the stalled invasion situation, and it relevant for entry into prey cells, it makes me wonder about the metallation state of Bd0875 and if it is binding the Ca^{2+} from solution instead of the putative Mg^{2+} listed. I find at least one report that MIDAS domains can bind Ca^{2+} . This needs to be discussed in more detail, and the argument about the function of the MIDAS domain would be strengthened with additional data from predation assays that explicitly consider metals and metal concentrations, such as metal chelators or supplemental metal.

<https://www.nature.com/articles/s41598-018-21231-1>

2B) The authors suggest that the additional cap region on Bd0875 is responsible/important for distinguishing Bd0875 from other MIDAS proteins. While I appreciate the approach to structurally model these proteins, this argument would be greatly strengthened by a chimeric approach of adding the cap region to another MIDAS protein and removing the cap region of Bd0875.

3) The authors attempt to determine the proteins secreted in the "kiss of death" was a clever use of gradient centrifugation. However, these results are only discussed briefly in the main text and

mainly relegated to the supplement. It would be useful to repeat this purification process at an earlier time point to try and capture some of the expected but missing targets. The language used to describe these results is somewhat ambiguous, leaving the reader somewhat confused if the results are novel. As in, if this proteomics experiment reveals proteins that are expected, is there a way to confirm these results? It currently reads as if any high-abundance peptides that aren't anticipated to be predatory proteins are reasoned away as being from attack cell lysis. For example, how are the authors sure that "non-predatory" peptides are in indeed non-predatory and not abundant proteins that might moonlight with roles in adhesion or invasion? Would it be possible to treat some subset of these purified samples with non-cell-permeable proteases to help remove contaminants that come from ruptured attack phase cells?

4) Minor: It would be helpful to see a Western blot of Bd0875-mCherry at T30, particularly of empty bdelloplasts, to confirm if the surface localization of the anti-mCherry spot is a cleavage product or still associated with a full length Bd0875.

Typographical/minor edits

5) Line 109 TFP as abbreviation is missing, my assumption, Type IV plus

6) Figure 3a, structures labels hard to read, had to zoom in a lot

We thank the reviewers for their work on, and interest in, our paper and for their clarification request which have improved our written explanations and the overall paper. We have replied in purple font and made changes in the manuscript with track changes for easy detection.

In The Data Availability Statement we have explained We have deposited all the full RNA Seq data to Genbank's Sequence Read Archive (SRA) (<https://submit.ncbi.nlm.nih.gov/about/sra/>) and they've been approved with accession numbers SAMN38260227- SAMN38260240. This has been included in the manuscript. The proteomic results are all included in the SOM of the paper and so are not deposited elsewhere.

REVIEWER COMMENTS

Reviewer #1 (Remarks to the Author):

The authors describe a very interesting phenotype in the predatory bacterium *Bdellovibrio bacteriovorus*. This work is significant for multiple topic areas, including bacterial predation, the overall roles of adhesins in bacteria, and evolution and ecology of bacterial lifestyles. This work advances our understanding of the role of specific proteins in the stages of *Bdellovibrio* predation, which is an important area of focus with potential practical applications against bacterial pathogens.

Thank-you for appreciating our results, you share our delight at the information from this mutant and its implications.

Please see specific comments below -

Line 19:

Bd0875..."important for efficient attachment to and successful invasion of prey"
The authors clearly demonstrated the role of Bd0875 in successful invasion of prey, but I'm not convinced about the role of Bd0875 in efficient attachment to prey. The experiments here focused of invasion, specifically the presence of empty but still rounded bdelloplasts as an indicator of failed invasions. To me, the current version of the manuscript doesn't support a conclusion about the role of Bd0875 in attachment.

Yes we understand this perspective which comes from the nature of the catch and release type adhesin of which Bd0875 is a family member. There isn't a measurable difference in attachment detectable in the presence/ absence of this protein in our paper for technical reasons. This is because in the frenetic motile arrival and invasion of predators at prey, the potential departure/unattachment, and fate, of 10% of *Bdellovibrio* is impossible to film even by timelapse (we did try) because there are a vast excess of additional predators in which any "failing to invade" predator cannot be traced microscopically.

However we clearly show that 10% of bdelloplasts do not have an invaded *Bdellovibrio* but we do find *Bdellovibrio* secreted proteins inside those 10% of uninvaded bdelloplasts and these are dead, despite the lack of invasion.

We inferred abnormal attachment as responsible for lack of invasion (which we could not measure, as above), but the reviewer is right that it could be prey to predator surface signalling of another sort, via the MIDAS adhesin protein abolishing invasion which we detected.

So to respond to this reviewer, we are removing "attachment" from the abstract at line 19 and have altered the abstract at line 24 and the discussion at **lines** 314-321, 335-341 to use abnormal "prey-interaction/ invasion" terms to avoid implying that abnormal attachment was measurable.

At original line 272, (**newline** 229) in the results we have replaced "early in the prey recognition and attachment process" with "early in the prey recognition and invasion process."

At original line 404 (**newline** 341) of the discussion we have replaced "prey invasions fail.. after a transient abnormal attachment by the predator", with "after a transient abnormal interaction by the predator". Also at original line 565 (**new line** 512) of the discussion we have HD100 proteins, secreted at the initial abortive attachment (around 15-30 minutes after predator and prey mixing) with HD100 proteins, secreted at the initial abortive invasion (around 15-30 minutes after predator and prey mixing)

We have

Thank you for making us spell out what could be measured versus inferred making this clearer in the paper.

.

Lines 36-38:

the dynamics of the initial interaction between predator and prey is a fascinating topic, but I'm not sure what this sentence is meant to convey about this interaction.

"marked change in movement-speed of the predator" - is this describing the predator's approach to a potential prey cell prior to attachment?

Yes you are correct that is the approach to a potential moving prey by a moving predator

"a struggle of associating and dissociating forces between the two initially live motile bacteria" - is this describing the physics of predators interfacing with prey, or molecular interactions between predator and prey outer surface components?

I suggest expanding explanations here and possibly including a citation.

Yes it is, sorry we didn't make that clear in shortform writing.

We have replaced lines 36-38 (**new line** 40) with the following to be clearer:-

The initial predator-prey cell encounter of two live moving bacteria becoming attached, brings a marked change in movement- speed of the predator. Also new associating and dissociating physical forces will occur between surface proteins as the predator interfaces with the initially-live prey cell surface

We have added the citation

Núñez ME, Martin MO, Duong LK, Ly E, Spain EM. Investigations into the life cycle of the bacterial predator *Bdellovibrio bacteriovorus* 109J at an interface by atomic force microscopy. *Biophys J*. 2003 May;84(5):3379-88. doi: 10.1016/S0006-3495(03)70061-7. PMID: 12719266; PMCID: PMC1302897.

Line 40:

"contracts" should be "contacts"? Yes sorry corrected in new wording at new line 40 as in answer to lines 36-38above

Line 74:

"license 'stick-and-slip' motility on host cell ligands"
I'm not sure what "license" means in this context

Thank you for asking us to write more to explain this more-

In this regard, "license" is used as dictionary definition – to allow/permit – but also with a subtext that this can be switchable *e.g.* is in a state beforehand that is not engaged but then becomes engaged to drive the resulting phenotype. It is often used when discussing the interplay of signalling and motility *e.g.* herein *Nat Rev Microbiol*. 2017 May;15(5):271-284. doi: 10.1038/nrmicro.2016.19.

In the motility for predation context: we know that Bd0875 is present on the surface of the invading *Bdellovibrio*, where motility engines including pili are involved in pulling the predator into the prey and where other adhesins are involved in making attachment contacts with host cell ligands. What we are explaining is that the Bd0875 is subject to movement forces that change its conformation and likely adjusts its binding activity as it experiences those forces.

We hypothesise that Bd0875 may be active as a kind of pivoting point as the *Bdellovibrio* enters the periplasm of the prey, powered by other engines and adhesins. That may require a kind of turn because the periplasm is narrow and surrounds the prey (as discussed in a recent paper about vibrio predator cell shape and prey entry by Banks et al). <https://doi.org/10.1038/s41467-022-29007-y>. We have added a sentence about this at new **line** 205 in the results.

Line 82:

"gave a significant percentage of failed predations"
are you considered these failed predations because the prey is killed but the predator does not replicate ?

We did consider this but we used fluorescently cytoplasmically tagged predators (Bd0064 mcherry or mcerulean) and never saw any evidence of prey-internalised fluorescence, or predator cell structures, inside prey empty bdelloplasts by microscopy, Hence we made this conclusion of failed predation.

does the predator gain any nutrients from killed but empty bdelloplasts?
what is the fate of the Bd0875 mutant predator that attaches, kills, but fails to invade?
The 10% of predators that did not invade were present in a vast excess of predators that were swimming in the vicinity of the prey, so it was experimentally/microscopically impossible to trace and follow predators that failed to invade. We presented what was possible to experiment upon.

Line 108:

what was the "parent strain" for the HI Bd1291 mutant? was it HD100? Yes it was we have added that in at original line 109, new **line 111**

Line 119:

"for B. bacteriovorus as Bd0875 has a lipobox" - delete "as"? No this means that having the lipobox is an indication of a surface role, we've added a comma before "as" to make this clearer.

Lines 304-306:

"AlphaFold models of other secretive candidates are strongly suggestive of many novel enzymatic activities/folds is this dataset"

support for this statement?

We have added context to this by rewording to add after this statement " (*i.e.* folds are predicted with confidence and/or conserved residues converge at putative active sites or binding pockets)". **new line 263-4**

Figure 2a:

the following information would be helpful to the reader either in the legend or near the first mention of this figure in the text -

what do the green numbers indicate? what do the blue stars indicate?

there are two regions marked with pink circles - does the longer one correspond to the cap, and the smaller correspond to the "small domain" shown in yellow in the structure overlay? or do both regions form the cap?

We are sorry this wasn't all present and have now included it in an extended legend to Figure 2. The green numbers indicate disulphide bond positions 1 bonded to 1, 2 bonded to 2 etc. The blue stars indicate conserved MIDAS domain amino-acids. The large pink circled region forms the cap and the smaller pink circled region forms a cap-adjacent domain. The small domain shown in 2b (pale yellow) is a composite of amino-acids 1-56 and 326-378.

Line 404:

"a significant proportion of prey invasions fail after a transient, abnormal attachment of the predator" - can you include or explain some data to show how attachment is transient and abnormal compared to wild-type?

The reviewer is correct. It was not possible to quantify experimentally the failing prey invasions down a microscope because the prey were surrounded by hundreds of *Bdellovibrio* all moving very quickly at 100+ microns per second. We tried making video movies but with only 10% of invasions failing and all predation events being surrounded by many other highly motile predators, we could not experimentally follow the fate of the single failing invasions that led to the empty bdelloplasts. As we mention earlier in this response, we have altered the wording to reflect this better.

a general comment:

is Bd0875 found in other intra-periplasmic *Bdellovibrio*? if so, are the key features conserved? Yes they are; we have added an alignment to Supporting Online Material at New SOM Fig. 4. We have also discussed this at **new lines** 314-321 where we also discuss the attachment and invasion point the reviewer made earlier.

Reviewer #2 (Remarks to the Author):

In this manuscript, Tyson and coworkers set out to determine the importance of a key step in predation of prey bacteria by *Bdellovibrio bacteriovorus*: prey entry. By combining proteomics, microscopy, and protein structural modeling, they were able to uncover a process they term the “kiss of death” where prey cells round up and die forming empty bdelloplasts, but the predator remains outside the prey cell. This process occurs in about 10% of bdelloplasts invaded by a strain of *B. bacteriovorus* lacking Bd0875. By synchronized imaging of predation, predator cells lacking Bd0875 attach to, but do not enter, the prey cells. The authors go on to examine the structural features of Bd0875 and structurally similar MIDAS proteins (Bd0767, Bd1483, and Bd3132). They use a genetic complementation approach to show that MIDAS domain was the key component to prevent empty bdelloplast formation. In addition, the authors were able to harvest empty bdelloplasts through density gradient centrifugation and determine that this “kiss of death” includes secretion of key predatory proteins. In general, the data presented mostly support the claims listed in the manuscript. I have included a few areas below where the authors should expand their experiments and some minor textual enhancements.

Thank you for this helpful and positive analysis of our paper.

1) Microscopy sampling and the stopping of predation. The authors should be complimented that they have such exquisite temporal control over their synchronized predation time courses.

Thank you. We have to have culturing and microscopy set up immediately adjacent to each other to manage to capture many of these images.

1A) It is confusing to me if the process of sampling 10 μ L of culture and immobilizing on a thin agarose pad is able to immediately halt predation. Given the dynamic nature of the process, the authors should comment on how they control for time since immobilization, how long it takes from the sampling of the predation flask to capturing the fluorescent images, and how long it takes to capture all the images.

The reviewer is correct that adding the liquid culture to prepared agarose slides does not halt predation, but imaging all the "views" for each sample takes less than 5 minutes and we controlled for this by imaging and saving the comparisons as wild type first in 2 experiments and the mutant first in 2 experiments. We then moved on to the next sampled timepoint for both cultures repeating the process as described.

All cultures and prepared slides were adjacent to the microscope in our lab to facilitate this speed with timers running. All the analysis was done after the event using Microbe J, so that fields of view were focussed and imaged as quickly as possible. We have added this to the methods in Section "Microscopy Analysis of Predation Timecourse" new line 412-419

1B) The images chosen for figure 1d portray a compelling story where the predator cell is able to enter the prey cell. However, I am not sure I am convinced that these images necessarily imply the predator has entered, and rather could be sitting on the surface of the bdelloplast. It may be that the authors know from personal experience, or different focal planes, or from dynamic/movies that the difference of internalized and attached but external are obvious, but this has not been presented clearly in this manuscript.

We tried to combine both dark predators in Fig 1d and fluorescently cytoplasmically labelled predators in Fig 1e to share the best views with readers that we could of these invasion processes.

The viewer has to remember that the invading fluorescent predators in Fig 1e are only invading underneath the outer-membrane (OM) of the prey, not inside the wall or cytoplasm at this stage, and the OM is a virtually transparent and this is why the Fig 1e look so bright. They are not sitting on the surface but are under a transparent OM cover only- We can see the invasion by eye down the microscope and that the predator is within the prey but there is little more we can use in proof (there are no facile reporters for OM engagement). We could remove Fig 1e if the reviewer thinks that Fig 1d is sufficient? We have added -("as fluorescent predator is entering under the transparent outer-membrane of prey"), to the legend for Figure 1e at newline 591. We have also added a typical image of a *Bdellovibrio* attached to the outside of a bdelloplasts as a contrast image in Fig 1e, see new Fig 1e.

2) Metal binding and MIDAS function.

2A) As Bd0875 expression is upregulated in in the stalled invasion situation, and it relevant for entry into prey cells, it makes me wonder about the metallation state of Bd0875 and if it is binding the Ca²⁺ from solution instead of the putative Mg²⁺ listed. I find at least one report that MIDAS domains can bing Ca²⁺. This needs to be discussed

in more detail, and the argument about the function of the MIDAS domain would be strengthened with additional data from predation assays that explicitly consider metals and metal concentrations, such as metal chelators or supplemental metal.

<https://www.nature.com/articles/s41598-018-21231-1>

We agree that metalation state of MIDAS proteins varies, and that particular members can even utilize Zn (<https://www.rcsb.org/structure/4cn9>). As we were unable to purify Bd0875, we were unable to demonstrate any preference. However, we do not believe that metal specificity of Bd0875 is behind any wider "*Bdellovibrio* whole cell properties", i.e. chelation or supplementation would have effects wider than the precise Bd0875 function/phenotype because several other proteins in or on *Bdellovibrio* predators may well respond to divalent ions. Thus, adding chelating agents to try and affect Bd0875 would interrupt these other proteins activity in an unseen way possibly affecting predation for other reasons. We didn't discuss this but could if it wasn't too confusing to add in.

2B) The authors suggest that the additional cap region on Bd0875 is responsible/important for distinguishing Bd0875 from other MIDAS proteins. While I appreciate the approach to structurally model these proteins, this argument would be greatly strengthened by a chimeric approach of adding the cap region to another MIDAS protein and removing the cap region of Bd0875.

Although the CAP represents a linear region of the Bd0875 fold (aa 143-176), it packs against two other loops (including the X residues of the DXSXS motif and also several hydrophobic amino-acids and that mean it acts like a co-operatively folded element). Therefore simple "cut-and-paste" chimeras are not possible. Likewise, it cannot be deleted/shortened as these complimentary hydrophobic amino-acids from elsewhere in the fold (at least 5 residues) would be exposed. This is why we experimentally used the single amino-acid site directed mutagenesis in the MIDAS motif but were unable to test with chimeras or cap region deletion as the structural model shows it would abolish overall protein packing and folding.

3) The authors attempt to determine the proteins secreted in the "kiss of death" was a clever use of gradient centrifugation. However, these results are only discussed briefly in the main text and mainly relegated to the supplement. It would be useful to repeat this purification process at an earlier time point to try and capture some of the expected but missing targets.

The language used to describe these results is somewhat ambiguous, leaving the reader somewhat confused if the results are novel.

We can't repeat it at an earlier time point because the predator-occupied bdelloplasts cannot be adequately separated from the empty ones. This is the experimental strategy and opportunity of what we discuss in this paper. We are only able to separate the empty bdelloplasts from the attack phase free *Bdellovibrio* predators as the buoyant density of occupied and unoccupied bdelloplasts isn't different enough. Thus, at earlier

timepoints there are too many occupied bdelloplasts; only at later timepoints are there only attack phase predators present, which can be separated from empty bdelloplasts due to their different buoyant density.

As in, if this proteomics experiment reveals proteins that are expected, is there a way to confirm these results? It currently reads as if any high-abundance peptides that aren't anticipated to be predatory proteins are reasoned away as being from attack cell lysis.

For example, how are the authors sure that "non-predatory" peptides are in indeed non-predatory and not abundant proteins that might moonlight with roles in adhesion or invasion? Would it be possible to treat some subset of these purified samples with non-cell-permeable proteases to help remove contaminants that come from ruptured attack phase cells?

We did not intend to imply or reason this at all, in fact only some of the "known from previous predation experimentation proteins" are still present in the empty bdelloplasts. It was just reassuring to see some of them present; because it implies that the "unknown" proteins with them are put into prey by predator at a relevant time to damage or kill the prey.

Indeed, it is the "not previously studied proteins" that are new and interesting and probably have unknown, novel predation roles worthy of further study by the community beyond our paper.

Because the Sockett lab has to close due to family healthcare requirements, this dataset is a set of proteins likely to be worth studying for attack phase activities and prey cell damage and we offer it to the community in this paper.

4) Minor: It would be helpful to see a Western blot of Bd0875-mCherry at T30, particularly of empty bdelloplasts, to confirm if the surface localization of the anti-mCherry spot is a cleavage product or still associated with a full length Bd0875.

Thank you. We have sequenced and verified the full length tagged genes amplified from the *Bdellovibrio*.

Typographical/minor edits

5) Line 109 TFP as abbreviation is missing, my assumption, Type IV plus YES, sorry added now just after TFP **new line** 112.

6) Figure 3a, structures labels hard to read, had to zoom in a lot.

We have enlarged the Figure 3a to include bigger labels and made Fig 3ai) for the whole structure and Fig 3aii) for the MIDAS domain to give them more space and make more readable.

Reviewer #1 (Remarks to the Author):

Thank you for revising the manuscript.

I appreciate your attention to addressing my concern regarding whether the data presented in the manuscript address Bd0875's role in attachment to prey. I recognize the technical and experimental constraints around observing attachment, and I don't think the manuscript needs to definitively demonstrate Bd0875's role in attachment, given the focus on invasion. I am satisfied with the changes and edits you made to clarify that the data cannot directly address attachment.

There are two places where I would recommend further clarifying your text.

(1) In response to my question about "license 'stick-and-slip' motility", you included some very helpful explanatory text regarding this and your hypothesis of how Bd0875 may be acting, but it appears that this is only reflected in a single line in the revised manuscript. I think including this text would benefit your readers by more clearly outlining your thinking.

(2) "failed predations" - my comments here were not about asking for more experiments, but asking for more clarity regarding how you are defining "failed predations". An epibiotic predator is still a successful predator even though it doesn't invade. What about a typically intra-periplasmic predator that doesn't invade, but can still kill a prey cell? Is it a failed predator? It's a failed invader, sure, but how are you judging failed predation? It sounds like your data cannot definitively show that the mutant predator was unable to replicate using resources from killed prey. If this is true, I recommend more cautious language in this statement from the Intro: "The bd0875 (and active MIDAS 84 motif point mutant strains) gave a significant percentage of failed predations." Here, I suggest changing "failed predations" to "failed invasions".

Reviewer #2 (Remarks to the Author):

This revised manuscript addresses many of the major issues that the reviewers presented in their previous reviews. I thank the authors for their textual changes and their clarity in responding to the reviews.

Reviewer 1 comments: I find the author's updated language to be clearer and more accurately describes their data.

Reviewer 2, comment 2A: I appreciate that interpreting these experiments in the presence of excess metals and/or metal chelators could be complex if there was an impact on invasion. On the other hand, the authors' model makes a clear prediction that there should be a role for metal binding. Given this clear prediction, I am hopeful that these types of experiments are included in future followup studies.

Reviewer 2, comment 2B: I appreciate the extended description of the difficulties in designing the potential chimera and why mutants were therefore limited to single amino-acid substitutions. While this description in the review process is sufficient for me, it may be helpful for the general reader to also have this information. I also concede that adding too many "short asides" to the discussion can dilute the main message that the authors are communicating.

REVIEWERS' COMMENTS

Reviewer #1 (Remarks to the Author):

Thank you for revising the manuscript.

I appreciate your attention to addressing my concern regarding whether the data presented in the manuscript address Bd0875's role in attachment to prey. I recognize the technical and experimental constraints around observing attachment, and I don't think the manuscript needs to definitively demonstrate Bd0875's role in attachment, given the focus on invasion. I am satisfied with the changes and edits you made to clarify that the data cannot directly address attachment.

There are two places where I would recommend further clarifying your text.

(1) In response to my question about "license 'stick-and-slip' motility", you included some very helpful explanatory text regarding this and your hypothesis of how Bd0875 may be acting, but it appears that this is only reflected in a single line in the revised manuscript. I think including this text would benefit your readers by more clearly outlining your thinking.

Thank you for your thoughtful care with the wording.

We have added a new definition at line 83 and in the discussion we have imported a shortened version of the explanation from the rebuttal into lines 337-345

(2) "failed predations" - my comments here were not about asking for more experiments, but asking for more clarity regarding how you are defining "failed predations". An epibiotic predator is still a successful predator even though it doesn't invade. What about a typically intra-periplasmic predator that doesn't invade, but can still kill a prey cell? Is it a failed predator? It's a failed invader, sure, but how are you judging failed predation? It sounds like your data cannot definitively show that the mutant predator was unable to replicate using resources from killed prey. If this is true, I recommend more cautious language in this statement from the Intro: "The bd0875 (and active MIDAS

84 motif point mutant strains) gave a significant percentage of failed predations." Here, I suggest changing "failed predations" to "failed invasions".

We have done this at new line 91.

Reviewer #2 (Remarks to the Author):

This revised manuscript addresses many of the major issues that the reviewers presented in their previous reviews. I thank the authors for their textual changes and their clarity in responding to the reviews.

Reviewer 1 comments: I find the author's updated language to be clearer and more accurately describes their data.

Reviewer 2, comment 2A: I appreciate that interpreting these experiments in the presence of excess metals and/or metal chelators could be complex if there was an impact on invasion. On the other hand, the authors' model makes a clear prediction that there should be a role for metal binding. Given this clear prediction, I am hopeful that these types of experiments are included in future followup studies.

Reviewer 2, comment 2B: I appreciate the extended description of the difficulties in **designing the potential chimera and why mutants were therefore limited to single amino-acid substitutions**. While this description in the review process is sufficient for me, it may be helpful for the general reader to also have this information. I also concede that adding too many "short asides" to the discussion can dilute the main message that the authors are communicating.

Thank you for your thoughtful care with the wording.

In the discussion we have imported a shortened version of the explanation from the rebuttal into lines 322-330.